# Real-Time Vehicle Classification and Tracking Using a Transfer Learning-Improved Deep Learning Network

**DOI:** 10.3390/s22103813

**Published:** 2022-05-18

**Authors:** Bipul Neupane, Teerayut Horanont, Jagannath Aryal

**Affiliations:** 1Advanced Geospatial Technology Research Unit, Sirindhorn International Institute of Technology, 131 Moo 5, Tiwanon Road, Bangkadi, Mueang Pathum Thani 12000, Pathum Thani, Thailand; geomat.bipul@siit.tu.ac.th; 2School of Information, Computer, and Communication Technology (ICT), Sirindhorn International Institute of Technology, 131 Moo 5, Tiwanon Road, Bangkadi, Mueang Pathum Thani 12000, Pathum Thani, Thailand; 3Department of Infrastructure Engineering, Faculty of Engineering and IT, The University of Melbourne, Melbourne, VIC 3010, Australia; jagannath.aryal@unimelb.edu.au

**Keywords:** vehicle detection, vehicle counting, vehicle classification, vehicle tracking, intelligent transport systems, multi-vehicle tracking

## Abstract

Accurate vehicle classification and tracking are increasingly important subjects for intelligent transport systems (ITSs) and for planning that utilizes precise location intelligence. Deep learning (DL) and computer vision are intelligent methods; however, accurate real-time classification and tracking come with problems. We tackle three prominent problems (P1, P2, and P3): the need for a large training dataset (P1), the domain-shift problem (P2), and coupling a real-time multi-vehicle tracking algorithm with DL (P3). To address P1, we created a training dataset of nearly 30,000 samples from existing cameras with seven classes of vehicles. To tackle P2, we trained and applied transfer learning-based fine-tuning on several state-of-the-art YOLO (You Only Look Once) networks. For P3, we propose a multi-vehicle tracking algorithm that obtains the per-lane count, classification, and speed of vehicles in real time. The experiments showed that accuracy doubled after fine-tuning (71% vs. up to 30%). Based on a comparison of four YOLO networks, coupling the YOLOv5-large network to our tracking algorithm provided a trade-off between overall accuracy (95% vs. up to 90%), loss (0.033 vs. up to 0.036), and model size (91.6 MB vs. up to 120.6 MB). The implications of these results are in spatial information management and sensing for intelligent transport planning.

## 1. Introduction

A fundamental source of the economic growth of any nation depends on well-planned and resilient transportation systems based on spatial information. Regardless, most cities around the world are still facing a rampant increase in traffic volume and complications in traffic management, resulting in poor quality of life in modern cities. However, recent advancements in internet bandwidth, artificial intelligence, and sensing technologies have minimized these difficulties by collaboratively bringing forward location intelligence for public safety. Automation in location intelligence in road environments using sensing technologies allows authorities to achieve resilience in road safety, controlled commutes, and assessments of road conditions [1]. This resiliency gives a positive direction towards achieving the United Nation (UN)’s Sustainable Development Goal (SDG) 11.2 [2].

Vehicle detection and classification using deep learning (DL) and multi-object tracking (MOT) on video streams obtained from a network of surveillance cameras have become the state of the art (SOTA) in the automation of intelligent transport systems (ITSs) [3]. These methods, when coupled with a fast-tracking algorithm, can provide real-time multiple vehicle tracking, depending upon the accuracy and speed of detection. However, the DL models need to be trained on a large training dataset and still can fail to produce accurate multi-object detection and classification. We used a network of existing surveillance cameras facilitated by the Department of Rural Roads (DRR) of Thailand fueled with SOTA DL-based models and propose a multi-vehicle tracking algorithm to achieve cost-effective, accurate, and real-time traffic monitoring systems. The major contribution of the research are as follows:We compared the existing SOTA networks of DL for real-time vehicle detection and classification.We minimized the effects of domain-shift that the DL networks suffer from with the help of transfer learning-based fine-tuning.We created and provide a vehicle classification dataset of seven vehicle classes.We propose a multi-vehicle tracking algorithm that obtains lane-based statistics such as count, speed, and average speed along with the vehicle classification.

The aforementioned contributions come from the overall methodology of our work, which tackles several prominent problems (P1, P2, and P3) of DL-based object classification. The problems are the requirements of a large labeled training dataset (P1), the domain-shift problem due to the difference between the training and testing data domain (P2), and coupling the real-time multi-vehicle tracking algorithm with the DL models. To tackle P1, as the existing training data mostly provided four classes of vehicles (car, bike, truck, and bus), we further divided these classes into seven classes including car, bus, taxi, bike, pickup, truck, and trailer in a newly proposed dataset of nearly 30,000 samples. We provide the dataset as a contribution to this research. For P2, to minimize the effects of domain-shift between two datasets, we trained the DL models in the large training data that we created and then fine-tuned the model again on a separate small training data that was collected from the cameras that were installed for implementation purposes. The fine-tuning was based on a transfer learning approach. Several SOTA DL models in the domain of near real-time multi-object classification belonging to the YOLO family [4] (two versions of improved YOLOv3 [5] and two versions of YOLOv5 [6]) were trained. The models were ensembled such that it tackles several of the existing challenges of real-time object detection, recognition, and classification. The challenges tackled by these YOLO models include the absence of an integrated anchor box selection process, time-taking space-to-depth conversion, the gradient descent problem, weak feature propagation, a large number of network parameters, and problems in the generalization of objects of different sizes and scales. To address P3, we propose a multi-vehicle tracking algorithm that takes the inputs from the fine-tuned DL model and provides the per-lane vehicle count, the speed of individual vehicles, and an average of these numbers along with the vehicle classification.

Section 2 presents the existing literature on the components of the research domain, Section 3 presents the overall method for the problem of real-time vehicle classification and tracking using the fine-tuned DL model, Section 4 demonstrates the experiments, validation, and critical discussions, and the paper concludes with future research directions in Section 5.

## 2. Related Works and Background

### 2.1. Vehicle Detection and Classification

Object detection is one of the fundamental problems of engineering, science, and technology research. This consists of accurately locating an object in space and time as well as classifying those objects that utilize computer vision technology. Advancements in machine learning and computer vision in the domain of vehicle detection and classification have produced several studies.

Chen et al. [7] used support vector machines (SVMs) for road vehicle classification based on colors and vehicle types using the Columbia Object Image Library (COIL) database. Changes in the color of vehicles due to the reflection of surface and strong sunlight affected the detection. Cao et al. [8] proposed a method based on the histogram orientation gradient (HOG) feature, which is not sensitive to illumination change and can better distinguish the appearance and shape of objects. This feature was used to train SVMs for detecting vehicles. A comparison between the HOG and Haar-like features for vehicle detection can be found in [9]. Uke et al. [10] proposed moving vehicle detection for traffic count measurement from a single camera implementing OpenCV development kits. Other methods such as scale-invariant feature transform (SIFT) progressed on robust features (SURF) methods and the use of 3D models [11] have also been investigated. The traditional methods produced faster detection but also generated more false detection. Furthermore, the problem of vehicle classification has not quite been addressed accurately by these methods.

The problems from illumination change and the shape of objects that the previous methods faced have been better addressed by the DL-based methods, with decreased false detection and increased accuracy in vehicle classification. Jung et al. [12] proposed vehicle classification and localization in traffic surveillance using ResNet50 for vehicle classification, adding dropping convolutional neural networks (DropCNN) with fine-tuning to improve classification performance. Zhuo et al. [13] introduced vehicle classification using a CNN named GoogleNet [14] for large-scale traffic surveillance. They pre-trained on the ILSVRC-2012 dataset and fine-tuned it with another vehicle dataset to achieve better accuracy. Even though these CNNs showed a good performance in object detection, they require heavy computation and are sensitive toward scale changes [15]. Most recently, networks referred to as YOLO have been widely used for vehicle detection and classification in real time, which are presented in the next section.

### 2.2. Vehicle Detection Using YOLO (You Only Look Once)

YOLO [4] started to look at object detection as a regression problem in a single neural network. The method then became the SOTA in the domain of object detection with overwhelming performance. With continuous upgrades, five generations of YOLO architecture have been produced since its introduction: YOLO, YOLOv2 [16], YOLOv3 [5], YOLOv4 [17], and YOLOv5 [6].

The first generation, YOLOv1, unified the methods of feature extraction, object localization, and classification to form a single-stage architecture. This network was SOTA in terms of mean average precision (mAP) with a fast detection speed. YOLO was a continuous series of convolutional layers with occasional maxpool layers at its time. Among other changes, YOLOv2 eliminated the fully-connected layer at the end of YOLOv1, allowing the network to perform independently of image resolution. The third generation, YOLOv3, continued with the fundamentals of its first two predecessors and was inspired by architectures such as ResNet [18] and the feature-pyramid network (FPN) [19]. Compared to other fast models such as Faster-RCNN [20], single shot multibox object detection (SSD) [21], and Center Net [22] on the COCO-2017 dataset, YOLOv3 is able to produce a similar mAP, but 17 times faster (see https://cv.gluon.ai/model_zoo/detection.html (accessed on 21 April 2022) for comparison).

Sang et al. [23] came up with an improved detection model based on YOLOv2, which used the k-means++ algorithm to train a dataset to cluster vehicle bounding boxes. To improve the loss due to different scales of vehicles, normalization was introduced. Moreover, repeated convolution layers were removed to improve feature extraction. Du et al. [24] proposed the real-time detection of vehicles and traffic lights with the YOLOv3 network, an improved version of YOLOv2, by detecting small objects with balanced speed and precision using a new, high-quality dataset named the Vehicle and Traffic Light Dataset (V-TLD). Song et al. [25] used YOLOv3 to detect and classify the vehicles and used an ORB algorithm [26] to obtain driving directions. Mahto et al. [27] used a fine-tuned YOLOv4 for vehicle detection using the UA-DETRAC dataset, which was faster than previous iterations. YOLOv5, despite being produced by a different author than its predecessors, has higher performance in terms of accuracy and speed among the YOLO family [6]. In our study, we explored the use of two variants each, YOLOv3 and YOLOv5, to form our methods. We did not use YOLOv4 because the YOLOv5 network we tested in our method design has a similar architecture and is smaller in model size. Studies related to vehicle tracking using these architectures are presented in the next section.

### 2.3. Vehicle Tracking

In this paper, we focus on single-camera tracking techniques for MOT, which allows for the detection of multiple objects in a video frame sequence. Performance in this type of tracking depends on the quality of detection and occlusion [28] as a single camera provides a one-sided view. The problem in detection due to these factors has been minimized by DL models that can detect objects with partial occlusion, meaning that, if the objects are not completely hidden by a larger object, the recent CNNs can still predict the object. Some of these DL networks that have been used in the real-time MOT include Faster-RCNN [20], SSD, and YOLO.

In the traditional tracking methods, the objects are first detected in the initial frames, and the feature is searched in the region to match them in a subsequent image sequence. Traditional detectors, such as contour-based target detection [29], the Harris corner detector [30], SIFT, and feature point-based methods [31,32], were used to produce false detection that resulted further in false matching. However, using DL models to detect the objects followed by matching features using the traditional tracking methods led to a higher performance. Bochinski et al. [33] tested several DL models to produce detection and used a passive detection filter for MOT, producing results in one of the fastest tracking methods. Zhang et al. [34] coupled the methods of Camshift [35] and the Kalman filter [36] along with a Faster-RCNN DL model [20] to detect and track vehicles in real time. The Kalman filter is commonly used to match features and track objects with high accuracy [37].

Following the tracking-by-detection techniques, Hou et al. [38] proposed the use of a method of tracking called DeepSORT with low-confidence track filtering, which reduced the false detection produced by the original DeepSORT algorithm. More recently, Liu et al., in [39], proposed 3-D constrained multiple kernels, facilitated with Kalman filtering, to track objects detected by a YOLOv3 network. These recent but sophisticated tracking algorithms have improved the accuracy of object tracking, but they require heavy computational power. Here, we propose a simple object-centroid tracking algorithm to track the detection provided by YOLO-based DL networks in multiple lanes of the road in real time. Furthermore, this study compares the use of two YOLO variants, YOLOv3 and YOLOv5, to obtain a real-time vehicle tracking method that can process multiple video streams with a single GPU, using multi-threading techniques [40].

### 2.4. The Domain-Shift Problem and Transfer Learning

Many of the studies that use DL models are pre-trained on a large training dataset such as COCO and KITTI [41], which include such classes as the car, bike, truck, and bus. For our purpose, we classify the vehicles into seven classes: car, bus, taxi, bike, pickup, truck, and trailer. As some of the classes in the existing datasets include many of our desired classes (e.g., pickup and SUV are classified as trucks in the COCO dataset), we are unable to use these datasets and the pre-trained models in our method. Therefore, we create a training dataset from several surveillance cameras set up by the DRR of Thailand. However, there is a greater challenge in training a DL model on a custom dataset. DL-based methods have the advantage of accurate detection and classification, but only if the training and test datasets are from the same or similar environment [42] such that they are not affected by contextual and spatial uncertainties [43]. In practical applications such as traffic surveillance, a network trained on a large training dataset collected from various sources can still underperform when trying to detect vehicles using new video streams from newly added cameras. This problem is often called the domain-shift problem, which occurs because machine learning methods including DL assume that the training and test datasets are produced in the same or similar environments. To address and minimize the effects of this problem, we used the method of transfer learning-based fine-tuning [44,45] to improve the performance of the YOLO networks that we trained on the large custom-designed training dataset, further reducing training time and thus increasing efficiency.

The method of transfer learning to improve vehicle detection using DL models has been investigated by others. Wang et al. [46] improved the performance of a custom-designed CNN, and Wang et al. [47] improved the performance of a deep belief network (DBN) using transfer learning. Similarly, Jo et al. [48] improved the performance of a GoogLeNet CNN, Nezafat et al. [49] improved the performance of a ResNet CNN to classify truck body types into further classes, and Zhang et al. [50] improved the performance of DenseNet, NasNet, Inception, VGG19, and MobileNet CNNs. Taormina et al. [45] used transfer learning-based fine-tuning techniques to train several pre-trained CNNs.

## 3. Method

The method introduces an algorithm based on a fine-tuned neural network to detect vehicles, classify the detected vehicles in real time into one of seven classes (car, bus, taxi, bike, pickup, truck, and trailer), count the classified vehicles on each lane on the road, track each vehicle’s movement, measure the speed of an individual vehicle, and calculate the average speed in different ranges of time.

To summarize the overall method, video streams are collected from road surveillance cameras using 4G/5G broadband technology. The video is streamed to a computing server with high processing power using a Real-Time Streaming Protocol (RTSP), where the stream is registered, and polygons are drawn to depict each road lane. Once the camera streams and road lanes are registered, the deep learning model trained beforehand then detects, classifies, and tracks the vehicles that enter the polygon of the lanes. This multi-vehicle tracking algorithm produces the count, classification, and calculated speed of each vehicle in each lane and then calculates the average speed in different ranges of times as well as the mobility of the vehicles according to the information from the road lanes. The step-by-step explanation of the method is shown in Figure 1 and presented hereafter in this section.

### 3.1. Training Dataset

The neural network was trained on two levels to minimize the effects of domain shift that neural networks face. Therefore, two training datasets were prepared as TR1 and TR2:TR1 consists of a large number of training samples collected from surveillance videos to train a base Model M1, which was trained for a high number of steps.TR2 consists of a small number of samples collected from the cameras used on the highway during the implementation of the overall system to train the fine-tuned Model M2 with the base knowledge transferred from M1.

This two-level training significantly reduced training time and minimized the effects of domain shift, producing a model that can perform with an optimized trade-off between classification accuracy and speed of detection.

For TR1, the road maintenance unit under the DRR of Thailand provided 6.3 TB of surveillance videos, taken from 23 cameras for 3 continuous days from 25 to 27 June 2020. Videos were selected such that we could prepare a training dataset that includes vehicle samples at different times of the day and different locations. The selected videos were then converted into image frames for annotation. We used an open-source program called *labelimg* for the task of manual annotation. Seven classes of vehicles, including car, bus, taxi, bike, pickup, truck, and trailer, were manually annotated for our purpose. As the sample of buses was not sufficient, we added 4431 samples of buses from a dataset provided by the authors in [25]. The total number of samples prepared for the TR1 dataset is shown in the “TR1 Samples” column in Table 1. Out of these total samples, 80%, 15%, and 5% were used for train, validation, and test datasets, respectively. The dataset is available as per the Data Availability Statement (https://github.com/bipulneupane/Thai-Vehicle-Classification-Dataset/ (accessed on 12 May 2022)).

For TR2, a smaller set of labels was prepared from the image frames collected from the cameras used for experimentation. Similar to before, *labelimg* was used to annotate the images with the same seven classes in the same order from car to trailer. However, this time, we included the 5% test samples from TR1, which were unused for training but used for visualizing results from M1. This addition of test data increased the number of training samples by re-using the previously unused samples. The number of these new labels and total samples after the addition of test data from TR1 are shown in the column “Labels for TR2” and “TR2 Samples” of Table 1. The ratio between the train, validation, and test datasets was kept the same. Some samples of the vehicle classes are shown in Figure 2.

### 3.2. YOLO-Based Detection and Classification

In this section, we describe the object detection models used in this study. The overall method was tested with two versions each of YOLOv3 and YOLOv5, ranging from the smallest to the largest model size, providing different trade-offs between speed and accuracy of detection. The key difference between these versions that make them different in size is the scaling multipliers of the width and depth of the network. For our purpose, we chose YOLOv3, YOLOv3-tiny (abbreviated as YOLOv3t), YOLOv5-large (abbreviated as YOLOv5l), and YOLOv5-small (abbreviated as YOLOv5s) for our comparison because these two versions allow us to compare the trade-off between accuracy and the speed of detection in real time without slowing down the overall method in our server setup.

In YOLOv3 (network architecture as shown in Figure 3), a DarkNet-53 backbone with batch normalization and leaky ReLU activation (abbreviated as DBL) is used without fully connected layers to extract features, firstly preparing feature maps from an image. A 13 × 13 grid is drawn over the feature map, and the object is predicted with three bounding boxes of different scales to predict the object, which are eventually merged. The box with the highest intersection over union (IoU) is the prediction. Small and large objects are detected using shallow and deep features, respectively, which allows the network to detect objects even if the scale changes. The DBL backbone in YOLOv3 uses a residual structure instead of the direct-connected CNN used by the previous versions of YOLO. This allows for the direct learning of residuals, simplifying the complexity of training with an improved accuracy of detection. The difference between YOLOv3 and YOLOv3t is that the first one uses three scales (13, 26, and 52) as shown in Figure 3, and YOLOv3t uses two scales (13 and 26) to predict the objects. Exclusive in the original YOLOv3, we automated the manual step of calculation of anchor box sizes, which is proposed in YOLOv5. Unlike in the originally proposed YOLOv3 models, this integrated automation allows end-to-end training of YOLOv3 and YOLOv3t models without having to separately calculate the anchor box sizes.

The YOLOv5 network consists of three components: a backbone, a neck, and a head. The backbone is a CNN that can produce image features of varying granularities. The neck is a series of different layers that mixes and combines the image features coming from the backbone to be passed forward for prediction. Finally, the head takes the features and the bounding box from the neck for the final class prediction steps. These three components can be combined wisely to produce different DL architectures for efficient prediction. For the first component of the backbone, we used a Focus layer and a Cross Stage Partial (CSP) network [51,52]. A Focus layer transforms the input images from space (resolution) to depth (number of channels). In classical models such as ResNet, a Conv2d layer uses a kernel and stride to reduce the space of the image and increase the depth. Unlike the Conv2d layer, a Focus layer reduces the computational expense by reshaping the tensors. The CSP network is based on DenseNet, which can effectively alleviate the problem of a vanishing gradient, strengthen the feature propagation, and reduce the number of parameters by reusing the existing features. CSP is therefore more efficient than commonly used ConvNet backbones of larger network parameters.

For the neck of YOLOv5, we used the CSP and a path aggregation network (PANet) [54] with a spatial pyramid pooling (SPP) block [55]. This neck generates feature pyramids that help the overall network to generalize objects of different scales and sizes. Using feature pyramids, PANet, and SPP blocks improves object identification on unseen data as well. Unlike our choice of PANet, the neck can be replaced with other networks, such as a feature pyramid network (FPN), a path aggregation network (PAN), and a bidirectional FPN (BiFPN). However, the creators of YOLOv5 found that PANet worked best for the YOLOv5 structure. In the head, the default YOLOv3 is used to apply anchor boxes on the generalized features to produce the final vector outputs with a bounding box, a probability of class prediction, and an objectness score. The network diagram is shown in Figure 4. The major advancement in YOLOv5 is that it integrates the anchor box selection process into the network, as it automatically learns the best anchor boxes for the training dataset. As mentioned before, we added this step into the training of YOLOv3 networks as well. The fundamental structure of YOLOv5l and YOLOv5s is the same. The difference is that the number of depths increases from the smaller to larger variants. The YOLOv5 networks convert space (image) to a lower number of depths, reducing the model size and accuracy while increasing the speed of detection.

To evaluate the YOLO networks, we used an improved intersection of union (IoU), i.e., the generalized intersection over union (GIoU) [56], for the loss function:(1)$$L_{\mathrm{GIoU}}\left(w\right)=1-\mathrm{IoU}+\frac{|C(A\cup B)|}{|C|}$$
where *A* and *B* are the bounding boxes of the ground truth and prediction, respectively, *C* is the smallest rectangle circumscribed between *A* and *B*, and IoU is the intersection of *A* and *B*. The major improvement of GIoU compared to IoU is that it defines a minimum closed area *C* such that the borders of *A* and *B* are included in *C*. GIoU then calculates the area of *A* and *B* not included in *C* proportionate to the total area of *C*.

All the trained networks were supplied with resized image frames of 640 × 640 size for both training and implementation. As mentioned before, the models were trained on the training dataset TR1 first and later on the TR2 dataset using transfer learning-based fine-tuning. This process is described in the next section.

### 3.3. Transfer Learning-Based Fine-Tuning

After the models were trained on TR1, these trained models could be used as a base model with a larger knowledge base. In the second phase of training, the same model architecture as before was trained on TR2, but with weights initialized from the previously trained model. This fine-tuning technique was an effective strategy for us to transfer and store knowledge gained by Model M1 from TR1 and apply it to Model M2 trained on TR2. The hyper-parameters and augmentation steps remained the same as before, but this time the model was trained for a significantly lower number of steps. This process produced models that produce higher performance when coupled with our tracking algorithm, as transfer learning from M1 to M2 effectively minimized the effects of the domain-shift problem that the neural networks face. In the next section, we present the step of the region of interest (ROI) selection, which reduces false detection.

### 3.4. Camera Registration and Road ROI Selection

The first step of the camera stream collection and registration was to obtain a real-time video stream from a surveillance camera on a highway. The camera was required to face the road lane with an angle of 40 to 50° from the nadir. The optimum height of the camera could be decided from the video streams that were used to produce TR2. The camera was connected to a 4G/5G broadband network to stream the video to a computer server that runs a monitoring system based on the trained YOLO model. The server can locate the camera stream with a unique RTSP link registered with the camera ID. Road lanes were manually drawn over an image frame of the stream such that the length of the sides of the polygon parallel to the flow of traffic was measured in meters to calculate the speed of vehicles. A lane ID was then given to the lane polygon, therefore completing the registration of the camera stream and lanes.

The lane polygons acted as a smaller ROI within the video frame, which prevented the false detection that mostly occurs on the horizons of the video, far from the camera. This increased the accuracy of the method, even if the training accuracy of the DL model was lower. Some studies have also segmented the road surface to achieve higher performance [25], which requires more computation work. The next step after registering the camera stream and lanes was to resize the video frames. This reduced the computational expense and reduced the loss of image data that may cause poor detection performance. We used the image size of 640 × 640, as the YOLO models were trained on this size to produce an optimum trade-off between accuracy and computation speed.

### 3.5. Multi-Vehicle Tracking with Speed Detection

The next step after training the YOLO models and collecting/registering camera streams was to use the trained model to track individual vehicle classes on the real-time video stream. For this, we developed a multi-vehicle tracking algorithm based on centroid tracking that takes the predicted class and bounding box from any DL model and performs several tasks to track vehicles, count the number of vehicles of each class, and calculate the speed in each lane polygon of the road. This method shows superior performance in terms of computational power, speed, and matching costs.

Describing the multi-vehicle tracking algorithm in Figure 1, the first step is to use the trained model to detect vehicle class and obtain a bounding box of the detected vehicles on the registered video streams. This bounding box and the lane polygon with its ID are passed into our multi-vehicle tracking algorithm. The following steps describe the algorithm:Calculate the centroid of the bounding box and check whether the centroid falls inside any registered lane polygons within the video stream.(a)If not, it is rejected/deregistered, meaning that the vehicle class and bounding box are stored, but do not pass through the tracking process.(b)If yes, the filtered vehicle is checked as to whether it matches any existing vehicle that was detected in previous image frames.If not (the vehicle is new), it is first registered with a new vehicle ID and passed to the step of updating features to the vehicle ID.If yes (vehicle exists), it is directly passed to the step of updating features to vehicle ID.Update the following features to vehicle ID: the ID of the vehicle, the lane the vehicle entered, the coordinates of the centroid of the vehicle on the image frame it first entered the lane polygon, the time of entry, the latest coordinates of the centroid of the vehicle, and the calculated speed of the vehicle. Calculate the following:(a)*Distance*: the difference between the coordinate of the latest and first centroid that entered the lane polygon. Convert distance to meters using the reference length provided earlier while registering the lane.(b)*Time*: the difference in time between when the vehicle was first recorded within the lane polygon and the current time recorded.(c)Speed := distance/timeRelease the vehicle ID as an existing vehicle to be matched with the upcoming vehicles.From the feature in Step 2, prepare a per-lane report including the count of class, the average speed in different ranges of time, and the vehicle mobility according to lane ID.

Now that the method has been explained in detail, in the next section, we will look at its performance and discuss the significant findings.

## 4. Results and Discussion

In this section, we describe the performance of the trained networks and the accuracy of the overall method. First, the training and validation accuracy of the YOLO models is compared upon the first and the second level of training (fine-tuning) to demonstrate the effects of domain-shift and how fine-tuning tackles the problem. The experimental results from the first level of training and its failure due to domain-shift is shown in Section 4.1, and the results after fine-tuning adjustments are shown in Section 4.2. Section 4.3 describes the results from coupling the fine-tuned YOLO models with our multi-vehicle tracking algorithm using four different cameras that show the vehicles from different angles. Several video streams taken at different times of day were used to validate the overall method for vehicle tracking in adverse lighting conditions.

For the purpose of comparison, we use metrics such as precision (P), recall (R), and mAP based on the precision/recall curve (PR-curve). The PR-curve is computed from the model’s confidence threshold. The recall is the proportion of all positive samples detected above the confidence threshold of 50%, and the precision is the proportion of all positive samples above the same confidence. P and R are calculated as shown in Equations (Equation 2) and (Equation 3):(2)$$\mathrm{Precision}\;\left(P\right)=\frac{\mathrm{TP}}{\mathrm{TP}+\mathrm{FP}}$$
(3)$$\mathrm{Recall}\;\left(R\right)=\frac{\mathrm{TP}}{\mathrm{TP}+\mathrm{FN}}$$
where TP, FP, and FN are the number of true positives, false positives, and false negatives, respectively. In the PR-curve, the precision at each recall level is interpolated by taking the maximum precision measured for a model for which the corresponding recall exceeds the recall level (*r*). The mAP describes the mean of average precision (AP) of the total number of classes (*n*), which in our case is 7. The AP summarizes the PR-curve and is defined as the average precision on a set of 11 equally spaced recall levels [0,0.1,…,1]:(4)$$\mathrm{AP}=\frac{1}{11}\sum_{r=0}^{1}p_{max}\left(r\right),r\in \left[0,0.1,\cdots ,1\right]$$
(5)$$\mathrm{mAP}=\frac{\sum \mathrm{AP}}{n}$$

### 4.1. Training YOLO Models on TR1

Here, we describe the training of YOLO models on the TR1 dataset and the results obtained. We wrapped the training codes in the Pytorch framework. Pre-trained weights were not used to initialize the training for the first level of training. Random data augmentation was applied during the training, which included a scaling of 0.5, a translation of 0.1, and a horizontal flip of 180 degrees. Moreover, the hue–saturation–value (HSV) was randomly changed with H, S, and V values of 0.015, 0.7, and 0.4, respectively. The input images were resized to 640 × 640 pixels when they were sent to the network. The initial learning rate was set to 0.01, and the final learning rate to 0.2, with a momentum of 0.937 and a weight decay of 0.0005. *Stochastic gradient descent (SGD)* was used as an optimizer to optimize the models. The models were trained in a batch size of 8 for 1000 epochs on a machine with 128 GB of RAM, an Intel(R) Xeon(R) Silver 4210 CPU, and two NVIDIA GeForce 2080 GPUs of 11 GB memory each. For all four networks, the anchor sizes were automatically learned and derived from the training dataset using a k-means clustering algorithm.

In the first level of training on TR1, models were trained for 1000 epochs. After the end of 1000 epochs, the best model was saved based on the highest mAP. The mAP with an IoU of 50 (i.e., mAP_50) and classification loss (class loss) on the training data of TR1 is shown in Figure 5. In our experiments, YOLOv5l produced the highest mAP_50, followed by YOLOv5s, YOLOv3, and YOLOv3t by the end of the 1000 epochs. The fifth-generation models produced a gradual increase in mAP, whereas the YOLOv3 models dropped in accuracy after around the 300th to 400th epoch. However, the classification loss of the YOLOv3 models seemed lower than the YOLOv5 models, with minimum loss shown by the YOLOv3 network. The overall loss values of all four models are below the acceptable range of 0.05.

For further comparison, AP and mAP were compared for the four YOLO networks trained on the TR1 dataset. Table 2 shows the distribution of the AP of the prediction vs. the validation dataset provided in the TR1 dataset for each of the seven classes of car, bus, taxi, bike, pickup, truck, and trailer. Even though the values fluctuate when compared among the individual classes, YOLOv5l shows the highest performance. The significant difference in training the four networks is in terms of training time. The fastest training model was YOLOv3t with the lowest mAP_50, and the slowest training model was YOLOv3 with the second-lowest mAP_50. The YOLOv5s and YOLOv5l, respectively, trained nearly four times and two times faster than YOLOv3. With the comparison, it was observed that the YOLOv5s model outperforms YOLOv3 in terms of both mAP and training time. YOLOv5l on the other hand shows the best trade-off among mAP_50, classification loss, and training speed.

The resulting sample outputs produced by the trained YOLOv5l model on the test images of TR1 are shown in Figure 6. It should be noted that the test samples in the image are irrespective of the ROI polygon, which was drawn during our tracking method. The ROI placed at a proper distance from the camera reduced false detection, resulting in a high accuracy of vehicle detection and classification, which will be shown later in Section 4.3. How the problem of domain-shift alters the accuracy of the trained models presented in this section will be explored in the next section.

### 4.2. The Domain-Shift Problem and Transfer Learning

In the previous section, it was shown that the four networks trained on the TR1 dataset produced a 74–80% mAP_50. However, due to the domain-shift problem that was described before, the same level of accuracy can only be expected on the image frames collected from the same camera as the TR1 dataset. Therefore, we tested the four TR1-trained networks on the TR2 dataset. In other words, all of the images and samples of TR2 were taken as test data for this experiment. The total number of samples in TR2 is shown in Table 1. The mAP_50 of all four models and AP for each class produced by the four models are shown in Table 3. This result highlights the domain-shift problem that many other research articles fail to demonstrate and represents the results from the models before fine-tuning adjustments. It can be seen that the mAP_50 of YOLOv3, YOLOv3t, YOLOv5l, and YOLOv5s reduced by 0.490, 0.494, 0.532, and 0.545, respectively.

To minimize the effects of domain shift, we fine-tuned the four TR1-trained YOLO networks into the TR2 dataset using transfer learning. During this fine-tuning (the second level of training), models were trained for 300 epochs, which is 700 epochs less than the first level of training. This short fine-tuning on a five-fold smaller dataset boosted the mAP of the model for all four networks (YOLOv3: 0.30 vs. 0.71; YOLOv3t: 0.26 vs. 0.67; YOLOv5l: 0.27 vs. 0.70; YOLOv5s: 0.25 vs. 0.68), as shown in Table 4, which represents the performance of the YOLO models after fine-tuning adjustments. The table also shows the classification loss, training time, and the size of the four trained networks. The classification loss seems to be under the desired threshold of 0.05; however, this time, the YOLOv3 version shows better performance than the YOLOv5 version. The reason behind this might be related to the graphs of mAP and classification loss shown in Figure 5. The YOLOv3 versions produced higher performance at the 300–400th epoch and gradually fell after that, whereas the YOLOv5 version showed a steady climb, beating the third-generation networks by the 1000th epoch. Looking at the model size, the newly trained models slightly increased in model size. However, the YOLOv5 versions still seemed to have a smaller size, which is crucial for our multi-vehicle tracking algorithm.

The fine-tuned models show a significant improvement in accuracy; however, there is more room for improvement in detection, counting, and tracking. It should be noted that the metrics are calculated based on the detection in a full image frame. The accuracy can be further improved if we limit the area of detection and avoid the objects far from the camera, where even the human eyes fail to correctly classify vehicles. Song et al. [25] improved the accuracy of detection and classification of cars, buses, and trucks from 49% to 95% by segmenting road pixels first, before object detection, using YOLOv3. Instead of segmenting the video stream using several image processing techniques with user-defined thresholds, we drew lane polygons over the video streams as described in Section 3.4. These lane polygons significantly minimized false detection in our multi-vehicle tracking algorithm, further allowing for speed calculation and the generation of lane-based reports for assessment. In the next section, we discuss the implementation and validation of our tracking algorithm.

### 4.3. Tracking, Counting, and Speed Detection from Fine-Tuned Networks

To implement the method and carry out our experiments, we used a setup of four cameras at a highway intersection in Ratchapruek, Pathum Thani, Thailand. The cameras were set up by the DRR at a height of 5.5 m with an angle of inclination of around 45 degrees in any direction, as shown in Figure 7. Broadband 4G internet was used to transmit the video streams for the experiments. Lanes were drawn with a distance between two parallel lines perpendicular to the flow of traffic of 10 m for speed calculation. Similarly, three other cameras were set up to experiment considering different angles. Camera 1 and 4 face the back and the front of the vehicles, respectively, and Camera 2 and 3 face on the sides. These cameras were the same as the ones used to prepare the TR2 dataset. To test the ability of our tracking algorithm, we tested it on all four cameras using all four YOLO models trained on TR2 with learning transferred from TR1. The same computer system used to train the networks was again used for these experiments.

The rate of the real-time stream was measured with the ratio of the frame per second (FPS) that the overall method could process to the FPS of the camera stream obtained through the RTSP. If the FPS of the stream and processing matches, then the processing is considered real-time. As we used two powerful GPUs, two camera streams were fed to one GPU using a multi-threading technique, providing two GPUs for four cameras. This helped in achieving real-time processing of the streams provided by the cameras. In the experiment carried out on four video streams collected from Camera 1–4, the maximum FPS that YOLOv3, YOLOv3t, YOLOv5l, and YOLOv5s could obtain when coupled with our tracking algorithm was 38, 64, 38, and 44, respectively. Depending on the model size, smaller models such as YOLOv3t and YOLOv5s could obtain a higher FPS. However, the input FPS from the RTSP was only 10 FPS, and all of the models were able to process the video streams in FPS higher than 10.

Table 5 shows the performance in detection and classification of the four YOLO networks trained on the TR2 dataset, tested on the four video streams. Out of all experiments, YOLOv5l produced the highest precision (P) (0.96) and recall (R) (0.95). The overall accuracy (OA), which is the average of P and R, was therefore 0.95. YOLOv3, YOLOv3t, and YOLOv5s had OA values of 0.90, 0.90, and 0.83, respectively. Unfortunately, the number of buses was quite low in the test videos, which explained the addition of training samples from another dataset into the TR1 dataset.

Some incorrect classifications from the use of YOLOv3 and YOLOv3s include confusion between pickups and cars, between trucks and trailers, between cars and taxis, and between vans (car in our case) and buses. Some pickups with a large load behind them were detected as trucks. Smaller vehicles hidden behind the large ones were missed. Smaller models such as YOLOv3t and YOLOv5s were unable to detect vehicles in all frames, resulting in an increased count in tracking methods. YOLOv5l on the other hand produced the least misses, increasing the performance in tracking.

It should be noted that the placement of lane polygons highly affected the accuracy of classification in our experiments. In one of the experiments using Camera 4, 10% more incorrect classifications were produced when the lane polygons were set up 10 m farther than shown in Figure 7. Setting the polygons further confused the networks among smaller vehicles such as cars, taxis, and pickups and large vehicles such as buses, trucks, and trailers. On the other hand, the proper placement of lane polygons minimized false detection.

In the next section, we discuss the performance of the so-far best-performing network, YOLOv5l, on the use of different quality videos.

### 4.4. Count and Classification on Different Qualities of the Stream

To show the performance of the method based on the image quality of the video stream, we provide stills from videos of different image qualities. Among the stills shown in Figure 8, the first one in Figure 8a (Vid 1) is from a video with noise, which is mostly due to the dust on the camera. The second (Figure 8b) and third (Figure 8c) stills (Vid 2 and Vid 3) were taken with a newly installed camera at 3 p.m. and 5 p.m. on the same day, respectively. The intensity of light at 3 p.m. is significantly higher than at 5 p.m., which is why they are included in this comparison. The fourth (Figure 8d) still (Vid 4) was taken at night at 3 a.m. The four experimental videos provided a practical environment for a comparison of how the quality of video affects the performance of our system. For the experiments in this section, we used the YOLOv5l network trained on the TR2 dataset, as it had been validated as the best performing network.

The result of the count using YOLOv5l on the four videos is shown in Table 6. It can be seen that, with a proper light intensity, the performance of vehicle detection and counting is most effective in the daytime with a clean, low-noise camera. At nighttime, the detection is least effective. There are several factors associated with the inefficacy of stream at nighttime, such as the black and white (B/W) image stream, the light from the headlights of vehicles, and the switching from B/W to color and from color to B/W during sudden changes in light intensity from a vehicle’s headlight. In overall, the average accuracy among the four videos was 94%.

For the validation of the classification task, we carried out our experiment on Vid 1 (noisy video) and Vid 3 (video with proper light intensity at 5 p.m.). As shown in Table 7, classification was best in the video with a higher light intensity. We did not include the performance at nighttime, because the classification was highly affected by several adverse factors. For nighttime, we plan to use a YOLOv5l model pre-trained on the COCO dataset to detect all vehicles as one class, without classifying them into seven classes.

## 5. Conclusions and Future Works

To achieve automation in location intelligence for public safety in road environments, this article presents an efficient and effective method of real-time vehicle detection, classification, and tracking using sensing technology. A new training dataset was first prepared to overcome the limitations of existing datasets and achieve domain- and country-specific precision in vehicle classification. We manually prepared nearly 30,000 samples with seven classes of vehicles (car, bus, taxi, bike, pickup, truck, and trailer) from Thailand. This dataset was used to train a state-of-the-art deep learning network to obtain a base model with a larger knowledge base. The base model was further fine-tuned to a new, smaller dataset prepared from the cameras that are ideal for the implementation of the system. This fine-tuning was achieved by using the transfer learning technique. The two-step process of training allows the model to leverage the learning from cross-domain datasets. Furthermore, the process reduces the number of training samples and iterations such that the model can be immediately implemented in real-world systems with minimized effects from the domain shift between the cross-domain datasets. The fine-tuned model was then combined with our multi-vehicle tracking algorithm, engineered to obtain the per-lane count, classification, and speed calculation of vehicles in real time.

The experimental training and fine-tuning were carried out with four deep learning models of the YOLO family: YOLOv3, YOLOv3t, YOLOv5l, and YOLOv5s. Among the four YOLO models, the architecture of the YOLOv5 models was assembled in our study using several components of existing networks to ensure the reduction of space-to-depth conversion time, alleviate the vanishing gradient problem, strengthen feature propagation, minimize the number of parameters by reusing existing features, and improve object identification on unseen data by improving the generalization of objects of different sizes and shapes. The four models were then tested to detect and classify the vehicles that utilize the algorithm that we developed. The YOLOv5l model obtained an overall accuracy of 95%. Moreover, in a different experiment, where the image quality was changed and the noise levels were varied, YOLOv5l could count and classify 99.6% and 94% of the vehicles, respectively, on a clear video stream.

In future work, we plan to validate the speed of the vehicles and compare the effects of Internet network speed, from 4G to 5G, on the use of DL models for road safety, and we also want to add license plate recognition (LPR) to our method. We also recommend testing different backbones and necks with the YOLOv5 architecture to increase the precision of location and vehicle classification.

## Figures and Tables

**Figure 1 sensors-22-03813-f001:**
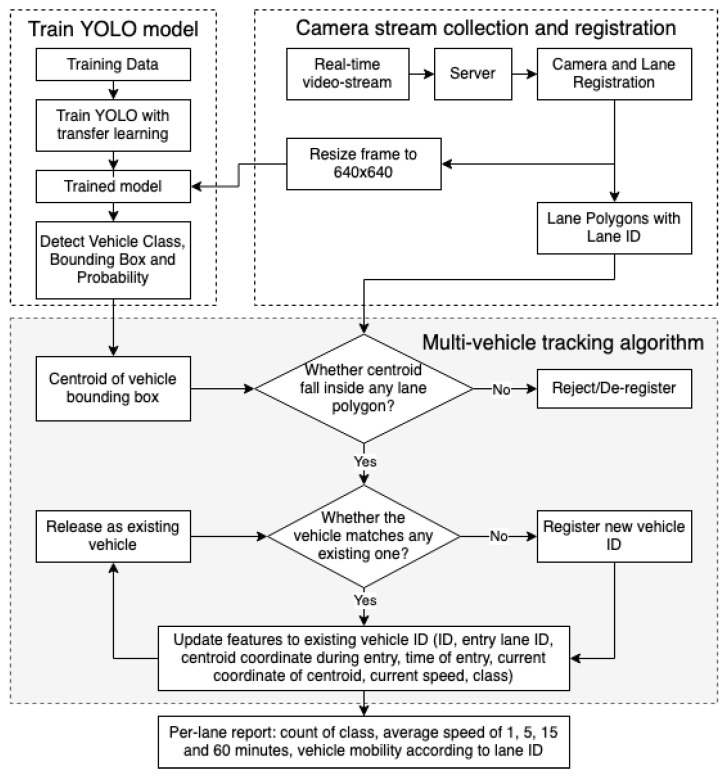
Multi-vehicle tracking algorithm for the lane-based count and speed detection of vehicles.

**Figure 2 sensors-22-03813-f002:**
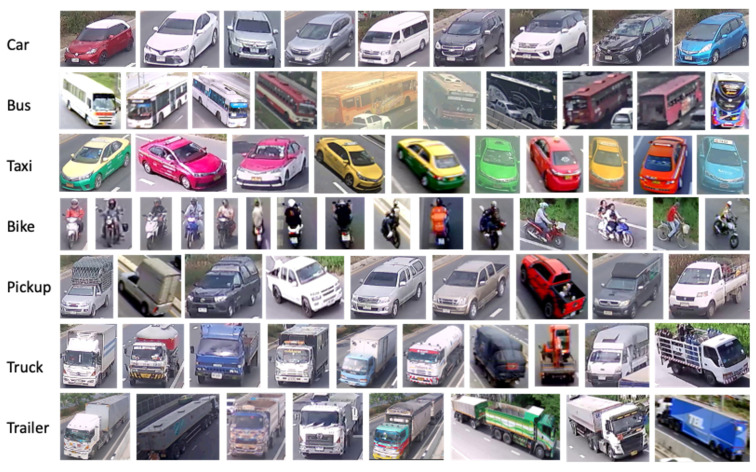
The samples of car, bus, taxi, bike, pickup, truck, and trailer (Classes 0 to 6).

**Figure 3 sensors-22-03813-f003:**
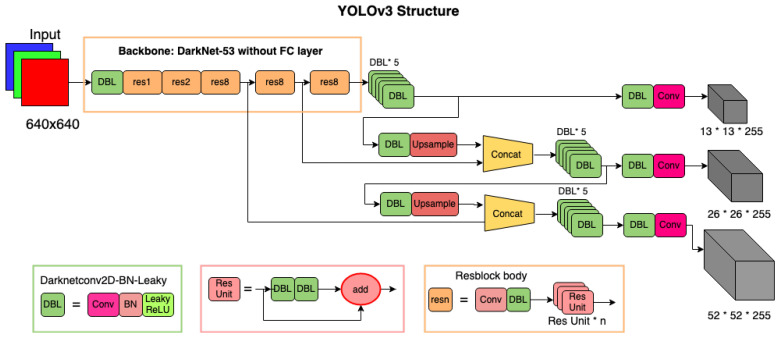
Network architecture of YOLOv3 (adapted from [53] and modified) with a backbone of DarkNet-53.

**Figure 4 sensors-22-03813-f004:**
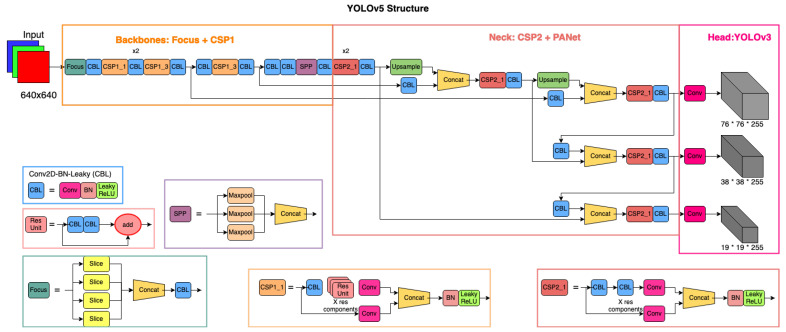
Network architecture of YOLOv5 (adapted from [57] and modified) with a backbone of Focus and CSP and a neck of CSP and PANet.

**Figure 5 sensors-22-03813-f005:**
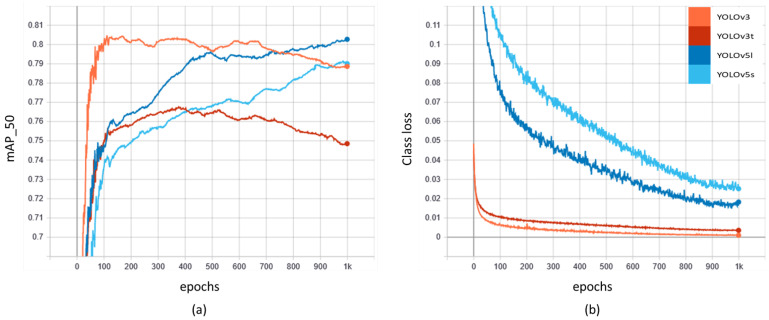
Network training results of YOLOv3, YOLOv3t, YOLOv5l, and YOLOv5s on the TR1 dataset. (**a**) mAP at IoU = 50; (**b**) classification loss.

**Figure 6 sensors-22-03813-f006:**
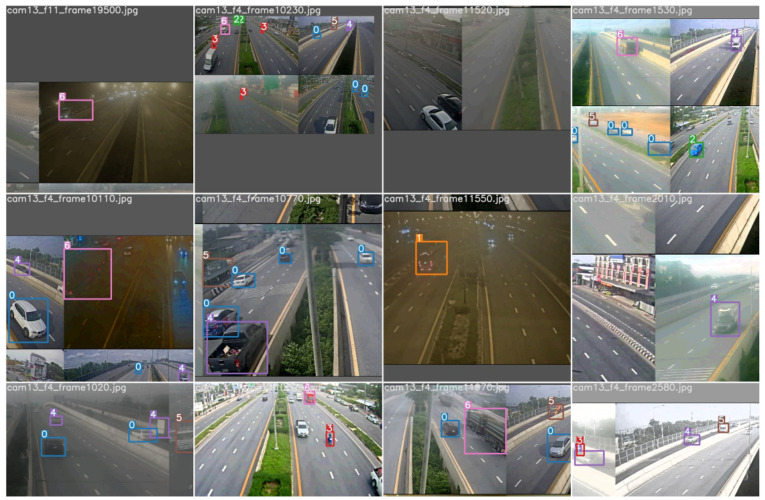
Sample results produced by the trained YOLOv5l model with the test images of TR1. The classes are predicted from 0 to 6, for car, bus, taxi, bike, pickup, truck, and trailer, respectively.

**Figure 7 sensors-22-03813-f007:**
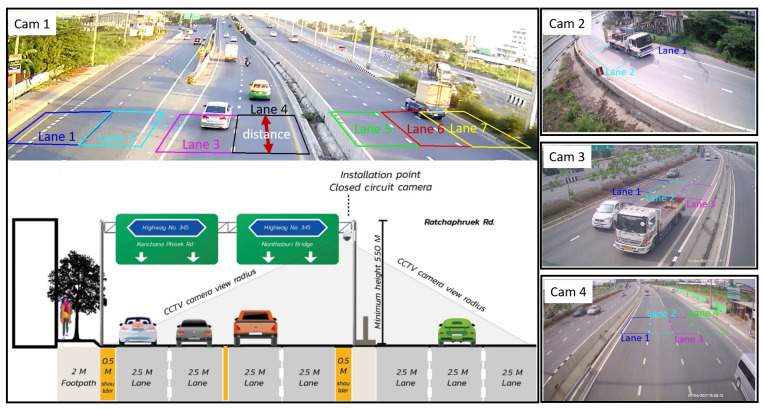
An experimental setup of four surveillance cameras for traffic monitoring. The lanes in the four streams are drawn as seen in their corresponding images (Cam 1, …, Cam 4).

**Figure 8 sensors-22-03813-f008:**
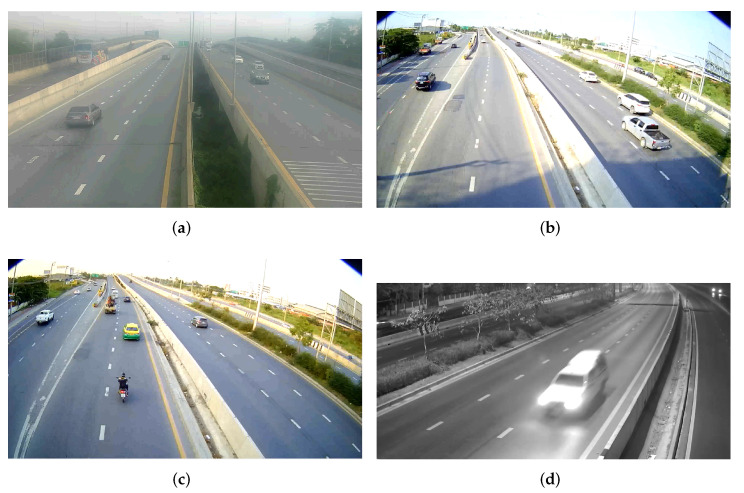
Sample image frames of four videos used for the experimentation on different qualities of streams. (**a**) Vid 1 (noisy video). (**b**) Vid 2 (high light intensity at 3 p.m.). (**c**) Vid 3 (proper light intensity at 5 p.m.). (**d**) Vid 4 (nighttime at 3 a.m.).

**Table 1 sensors-22-03813-t001:** Total number of samples in training datasets TR1 and TR2.

Vehicle Type	TR1 Samples	Labels for TR2	TR2 Samples
Car	10,478	2276	2585
Bus	540 + 4431	20	274
Taxi	1605	232	323
Bike	2572	508	562
Pickup	6056	1868	2042
Truck	2656	524	666
Trailer	1179	81	160
Total	29,474	5509	6612

**Table 2 sensors-22-03813-t002:** Performance of the YOLO networks trained on TR1 evaluated on the validation data of TR1.

Classes	Average Precision (AP)			
	YOLOv3	YOLOv3t	YOLOv5l	YOLOv5s
car	0.770	0.739	0.797	0.790
bus	0.933	0.906	0.926	0.913
taxi	0.765	0.788	0.846	0.849
bike	0.775	0.694	0.780	0.773
pickup	0.726	0.647	0.733	0.674
truck	0.849	0.819	0.828	0.841
trailer	0.703	0.648	0.708	0.690
mAP_50	0.789	0.749	**0.803**	0.790
Class Loss	0.001	0.003	0.018	0.025
Train time (hours)	101.03	35.03	65.03	37.01
Model Size (MB)	120.56	17.02	91.58	14.07

**Table 3 sensors-22-03813-t003:** Performance of the YOLO networks trained on TR1 evaluated on TR2 as test data before fine-tuning.

Classes	Average Precision (AP)			
	YOLOv3	YOLOv3t	YOLOv5l	YOLOv5s
car	0.314	0.338	0.326	0.369
bus	0.22	0.172	0.158	0.16
taxi	0.538	0.468	0.536	0.503
bike	0.338	0.258	0.358	0.304
pickup	0.296	0.268	0.279	0.284
truck	0.255	0.178	0.199	0.071
trailer	0.129	0.105	0.041	0.023
mAP_50	0.299	0.255	0.271	0.245

**Table 4 sensors-22-03813-t004:** Performance of the YOLO networks previously trained on TR1, fine-tuned on TR2, and evaluated on the validation data of TR2.

Classes	Average Precision (AP)			
	YOLOv3	YOLOv3t	YOLOv5l	YOLOv5s
car	0.764	0.755	0.722	0.745
bus	0.861	0.761	0.762	0.750
taxi	0.660	0.687	0.733	0.673
bike	0.792	0.724	0.817	0.784
pickup	0.757	0.719	0.686	0.711
truck	0.688	0.584	0.654	0.628
trailer	0.447	0.457	0.499	0.492
mAP_50	0.710	0.670	0.695	0.683
Class Loss	0.015	0.028	0.033	0.036
Train time (hours)	7.83	2.83	7.98	2.55
Model Size (MB)	120.59	17.02	91.61	14.09

**Table 5 sensors-22-03813-t005:** Performance of the YOLO networks trained on TR2 evaluated on validation video streams collected from the test area.

Camera	No. of Frames	Vehicle Face	Class	GT	YOLOv3			YOLOv3t			YOLOv5l			YOLOv5s		
					P	R	OA	P	R	OA	P	R	OA	P	R	OA
**Cam 1**	7500	Back	**car**	217	0.99	0.95	0.97	0.96	0.91	0.93	0.99	0.97	0.98	0.98	0.91	0.95
			**bus**	5	0.56	1.00	0.78	0.80	0.80	0.80	1.00	0.80	0.90	0.80	0.80	0.80
			**taxi**	6	0.86	1.00	0.93	0.86	1.00	0.93	0.75	1.00	0.88	0.86	1.00	0.93
			**bike**	20	1.00	1.00	1.00	1.00	1.00	1.00	0.95	1.00	0.98	1.00	0.90	0.95
			**pickup**	99	0.98	0.86	0.92	0.92	0.86	0.89	0.98	0.96	0.97	0.88	0.84	0.86
			**truck**	4	0.33	0.50	0.42	0.43	0.75	0.59	0.60	0.75	0.68	0.43	0.75	0.59
			**trailer**	1	0.50	1.00	0.75	0.50	1.00	0.75	1.00	1.00	1.00	0.00	0.00	0.00
			**Total**	352	0.74	0.90	0.82	0.78	0.90	0.84	0.90	0.93	0.91	0.71	0.74	0.72
**Cam 2**	9060	Side	**car**	16	1.00	0.88	0.94	1.00	0.94	0.97	1.00	0.94	0.97	0.93	0.88	0.90
			**bike**	1	1.00	1.00	1.00	1.00	1.00	1.00	1.00	1.00	1.00	1.00	1.00	1.00
			**pickup**	12	0.85	0.92	0.88	0.92	0.92	0.92	0.92	1.00	0.96	0.83	0.83	0.83
			**truck**	3	0.75	1.00	0.88	0.75	1.00	0.88	1.00	1.00	1.00	0.43	1.00	0.71
			**trailer**	4	1.00	1.00	1.00	1.00	1.00	1.00	1.00	1.00	1.00	1.00	0.25	0.63
			**Total**	36	0.92	0.96	0.94	0.93	0.97	0.95	0.98	0.99	0.99	0.84	0.79	0.82
**Cam 3**	8800	Side	**car**	87	0.96	0.93	0.95	0.99	0.86	0.92	0.98	0.94	0.96	0.99	0.83	0.91
			**taxi**	6	1.00	1.00	1.00	1.00	1.00	1.00	1.00	1.00	1.00	1.00	1.00	1.00
			**bike**	12	1.00	0.83	0.92	1.00	0.83	0.92	1.00	0.83	0.92	1.00	0.83	0.92
			**pickup**	91	0.93	0.93	0.93	0.87	0.97	0.92	0.96	0.98	0.97	0.86	1.00	0.93
			**truck**	12	1.00	0.58	0.79	0.86	0.50	0.68	0.92	1.00	0.96	0.82	0.75	0.78
			**trailer**	3	0.43	1.00	0.71	0.43	1.00	0.71	1.00	0.67	0.83	0.67	0.33	0.50
			**Total**	211	0.89	0.88	0.88	0.86	0.86	0.86	0.98	0.90	0.94	0.89	0.79	0.84
**Cam 4**	9575	Front	**car**	149	0.96	0.89	0.92	0.88	0.92	0.90	0.95	0.97	0.96	0.92	0.95	0.94
			**taxi**	4	1.00	1.00	1.00	1.00	1.00	1.00	1.00	1.00	1.00	1.00	1.00	1.00
			**bike**	25	1.00	0.96	0.98	1.00	0.96	0.98	1.00	1.00	1.00	0.96	1.00	0.98
			**pickup**	73	0.81	0.92	0.86	0.82	0.75	0.79	0.94	0.90	0.92	0.91	0.84	0.87
			**truck**	5	1.00	1.00	1.00	1.00	1.00	1.00	1.00	1.00	1.00	1.00	1.00	1.00
			**Total**	256	0.95	0.95	0.95	0.94	0.93	0.93	0.98	0.98	0.98	0.96	0.96	0.96
**Grand Total**				855	0.88	0.92	0.90	0.88	0.92	0.90	**0.96**	**0.95**	**0.95**	0.85	0.82	0.83

**Table 6 sensors-22-03813-t006:** Results of the count on different qualities of videos using YOLOv5l.

Video	Frames	Manual Count	Method Count	OA
Vid 1	90,000	3617	3390	93.7
Vid 2	7500	552	545	98.7
Vid 3	7500	514	512	99.6
Vid 4	7500	14	12	85.7
Average Acc.	94.4			

**Table 7 sensors-22-03813-t007:** Results of classification using YOLOv5l in terms of the quality of the video.

Video	Frames	Correct Classif.	False Classif.	R	P	OA
Vid 1 (Noisy)	90,000	2851	766	78.8	84.1	81.4
Vid 3 (Clear)	15,000	470	30	94.0	94.0	94.0

## Data Availability

The Thai Vehicle Classification dataset with 30,000 samples of vehicles is available on Github at https://github.com/bipulneupane/Thai-Vehicle-Classification-Dataset/ (accessed on 12 May 2022).

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
