# Peer review of "Real-Time Vehicle Classification and Tracking Using a Transfer Learning-Improved Deep Learning Network"

_sensors, 2022, doi:10.3390/s22103813_

Round 1
Reviewer 1 Report
This paper proposes a real-time vehicle classification and tracking approach using transfer learning-improved deep learning network. In general, the contents are interesting and implementable. Please consider the following comments:
1. There is some repetition between the divided chapters in the article, which should be introduced in detail.
2. The innovation of the article should be introduced in more detail and highlighted.
3. The pictures should not exceed the text area.
4. The improved Yolo algorithm should be described in detail, such as the structure.
5. Some of the contents mentioned in the keywords are not specified.
6. The comparison between the model after fine adjustment and the model before adjustment can be described in detail.
7. The different structures of the four models selected in the article should be described as a comparison.
8. There should be a reasonable explanation for the differences between the experiments made by the article selection model.
Author Response
Thank you for your overall positive comments and constructive criticisms. We have addressed the comments in the current version of the manuscript. The response to the comments is given below. Other than the response listed below, the major changes have been highlighted in the track changes document with green colour and the grammatical changes are highlighted with the blue colour.
Comments and replies:
This paper proposes a real-time vehicle classification and tracking approach using transfer learning-improved deep learning network. In general, the contents are interesting and implementable. Please consider the following comments:
C1. There is some repetition between the divided chapters in the article, which should be introduced in detail.
R1: Thank you for the comment. We would like to emphasize the idea which is important for the research community that might have been observed the repetition by the reviewer. However, the repetition is thoroughly examined in various chapters, and the repeated information is now deleted.
Example (deletion): We deleted the “YOLOv3 uses a backbone of DarkNet-53” in line 245 (of the previous version) as it is already mentioned at the beginning of the same paragraph. We replace this deleted writing by adding more explanation on how the backbone of DarkNet-53 replaces the CNNs used by previous YOLO versions (current line 265-266).
Example (repetition): (i) We have re-written the abstract to be short and precise. (ii) In the added paragraph of the introduction (lines 46-68), we provide the overall methodology to showcase and highlight the importance of our work.
C2. The innovation of the article should be introduced in more detail and highlighted.
R2: Thank you for this comment. The innovation of the article is now presented by adding paragraph 3 (lines 46-68) in the introduction (section 1).
C3. The pictures should not exceed the text area.
R3: The figures are generated using the most recent latex template provided by MDPI. The pictures that exceed the text area are the wide figures. Wide figure options are provided in the template as the guideline for authors. However, we will check with the Editor and follow their advice in laying out the figures.
C4. The improved Yolo algorithm should be described in detail, such as the structure.
R4. Firstly, thank you for this comment. We improve and describe the structure of Yolo algorithms - YOLOv3 and YOLOv5 in detail as below.
The YOLOv3 and YOLOv5 structures are explained in Section 3.2 (beginning from para 2 of this section). The improvement on the YOLOv3 on our part is the automation of the anchor box selection process and integration of the process within the YOLOv3 structure such that end-to-end training can be performed. Line 270 – 273 has been added to provide clarity to this improvisation. We also created and added a network diagram of the YOLOv3 structure in Figure 3. YOLOv3 and YOLOv3t are similar in structure. The difference is in the number of scales of the bounding box. YOLOv3 uses three scales of sizes 13, 26, and 52 and YOLOv3t uses only two scales of 13 and 26. We added lines 268-270 to explain this.
Similarly, in the case of YOLOv5, we have explained the structure in Figure 4 and provided the details in para 3, 4, and 5 of section 3.2. The components of YOLOv5 are chosen from the available options provided by the authors of YOLOv5 themselves. We do not add any new components. However, we choose the backbone of the Focus layer and Cross Stage Partial (CSP) and neck of CSP, Path Aggregation Network (PANet), and Spatial Pyramid Pooling (SPP) to ensure the reduction of space-to-depth conversion time, alleviate the vanishing gradient problem, strengthen the feature propagation, minimize the number of parameters by reusing the existing features, and to improve object identification on unseen data by improving the generalization of objects at different size and shapes. The neck could be replaced by a feature pyramid network (FPN), path aggregation network (PAN), Bidirectional FPN (BiFPN), etc. Paragraph 3 and 4 of section 3.2 describes this in detail. We added lines 293-296 to highlight the other options for the neck.
C5. Some of the contents mentioned in the keywords are not specified.
R5. Thank you for pointing this out. We have removed the “highway management” keyword that did not fit well with this article and replaced it with “intelligent transport system”.
C6. The comparison between the model after fine adjustment and the model before an adjustment can be described in detail.
R6. Thank you for this comment. To compare the models before and after fine-tuning, we have re-written some sentences and added more details (lines 467-470) in section 4.2. We edited the caption of Tables 3 and 4 as well. Table 3 represents the performance of the model before fine adjustment and Table 4 represents the performance of the model after fine adjustment.
C7. The different structures of the four models selected in the article should be described as a comparison.
R7: Figure 4 is added to show the network architecture of YOLOv3 versions. YOLOv3 and YOLOv3t are similar in structure. The difference is in the number of scales of the bounding box. YOLOv3 uses three scales of sizes 13, 26, and 52 and YOLOv3t uses only two scales of 13 and 26. We added lines 268-270 to explain this. In the case of YOLOv5, the structure is explained in Figure 5 and the explanation is added in paragraphs 3 and 4 of section 3.2. The fundamental structure of YOLOv5l and YOLOv5s is the same. The difference is that the number of depths increases from the smaller to larger variants. The YOLOv5s convert space (image) to a lower number of depths, reducing the model size and accuracy while increasing the speed of detection. This explanation is added in lines 302-305.
C8. There should be a reasonable explanation for the differences between the experiments made by the article selection model.
R8: This point is considered positively, and we added an explanation in the first paragraph of section 4 (lines 389-397) for experimental design with a focus on the difference between the experiments.
Reviewer 2 Report
Dear Authors,
The article concerns intelligent transport systems that play a very important role in road transport. The authors presented interesting studies of road traffic volume measurement based on deep learning model. The article is well-worded, but there are some shortcomings, which I highlight below.
1. First, it is too long to abstract, should be 200 words long.
2. The introductory part is well designed and gives a good overview of the solutions to the subject of the manuscript.
3. Section 3 Methods is well described.
4. The format of the tables should be corrected according to the guidelines for authors.
5. Figure 5 should enlarge.
The conclusions of the work reflect the research carried out. The authors also indicated further stages of work on the presented solution. Appropriate literature was used, which testifies well to the authors' writing skills.
In general, the manuscript is well written and there are some shortcomings, so I recommend the paper to be printed with minor corrections.
Author Response
Thank you for your overall positive comments and constructive criticisms. We have addressed the comments in the current version of the manuscript. The response to the comments is given below. Other than the response listed below, the major changes have been highlighted in the track changes document in green colour and the grammatical changes are highlighted in blue colour.
Comments and replies:
The article concerns intelligent transport systems that play a very important role in road transport. The authors presented interesting studies of road traffic volume measurement based on deep learning model. The article is well-worded, but there are some shortcomings, which I highlight below.
C1. First, it is too long to abstract, should be 200 words long.
R1: The abstract has been re-written to 200 words.
Previous:
Accurate detection, classification, and tracking of vehicles on video streams is an increasingly important subject for intelligent transport systems (ITS) and planning utilising precise location intelligence. Deep learning and computer vision provide effective methods, however, improving precision in real-time monitoring is a challenge and comes with problems. Three important problems are studied in this research- (i) the need for a large training dataset, (ii) domain-shift problem and performance issues due to the difference in training and testing environment, and (iii) coupling multi-object tracking algorithm with the deep learning model for real-time tracking. To address the first problem, we create a training dataset of nearly 30,000 samples using an existing network of surveillance cameras that includes more classes of vehicles than in existing datasets. For the second problem, we train and compare several state-of-the-art deep learning networks of the YOLO (You Only Look Once) family and further increase the accuracy by fine-tuning the models using transfer learning. For the third problem, we propose a multi-vehicle tracking algorithm to obtain the per-lane count, classification, and speed calculation of vehicles in real-time. The results show that the accuracy of fine-tuned models is more than double the models that suffered from domain-shift problems (71\% Vs up to 30\%). When coupled with our tracking algorithm, the YOLOv5-large model provides a trade-off between overall accuracy (95\% Vs up to 90\%) in detection and classification of vehicles, loss (0.033 Vs up to 0.036), and model size (91.6MB Vs up to 120.6MB) when compared among the YOLO family (YOLOv3, YOLOv3-tiny, YOLOv5-small, and YOLOv5-large models). The implications of these results are in spatial information management and sensing for intelligent transport planning.
Updated:
Accurate vehicle classification and tracking is an increasingly important subject for intelligent transport systems (ITS) and planning utilizing precise location intelligence. Deep learning (DL) and computer vision are intelligent methods, however, accurate real-time classification and tracking come with problems. We tackle three prominent problems (P1, P2, and P3): the need for a large training dataset (P1), domain-shift problem (P2), and coupling real-time multi-vehicle tracking algorithm with the DL (P3). To address P1, we create a training dataset of nearly 30,000 samples from existing cameras for seven classes of vehicles. To tackle P2, we train and apply transfer learning-based fine-tuning on several state-of-the-art YOLO (You Only Look Once) networks. For P3, we propose a multi-vehicle tracking algorithm to obtain the per-lane count, classification, and speed of vehicles in real-time. The experiments show accuracy has doubled after fine-tuning (71% Vs up to 30%). Coupling the YOLOv5-large network to our tracking algorithm provides a trade-off between overall accuracy (95% Vs up to 90%), loss (0.033 Vs up to 0.036), and model size (91.6MB Vs up to 120.6MB) when compared among four YOLO networks. The implications of these results are in spatial information management and sensing for intelligent transport planning.
C2. The introductory part is well designed and gives a good overview of the solutions to the subject of the manuscript.
R2: Thank you for your comment.
C3. Section 3 Methods is well described.
R3: Thank you for your comment.
C4. The format of the tables should be corrected according to the guidelines for authors.
R4: The tables are generated using the most recent latex template provided by MDPI. However, we will check with the Editor and use their help if necessary.
C5. Figure 5 should enlarge. The conclusions of the work reflect the research carried out. The authors also indicated further stages of work on the presented solution. Appropriate literature was used, which testifies well to the authors' writing skills.
In general, the manuscript is well written and there are some shortcomings, so I recommend the paper to be printed with minor corrections.
R5: Figure 5 has been enlarged. Thank you again for your overall positive comments and constructive criticisms.
Reviewer 3 Report
The article indeed looks promising in terms of idea that authors are presenting. However, I have some serious concerns that must be resolved before acceptance.
- The writing style, use of English langauge, and the grammer is very poor. You need to correct it thorougly.
- There are various other deep learning based approaches available for vechicle detection and classification. Why do you think that your approach should be adopted instead of exisiting approaches that are available.
- I think that vehicle detection approaches are not really important at present time because we already have such systems to do such tasks. But if your approach can also detect the vehicle number along with the type of vehicle then it can be better approach. Maybe you would like to consider this in the revised version.
Author Response
Thank you for your overall positive comments and constructive criticisms. We have addressed the comments in the current version of the manuscript. The response to the comments is given below. Other than the response listed below, the major changes have been highlighted in the track changes document in green colour and the grammatical changes are highlighted in blue colour.
Comments and replies:
C1. The writing style, use of English language, and the grammar is very poor. You need to correct it thoroughly.
R1: Thank you for your comment. We have revised the grammatical errors and improved the level of writing using Grammarly. The changes are shown by blue highlights in the track changes document.
C2. There are various other deep learning-based approaches available for vehicle detection and classification. Why do you think that your approach should be adopted instead of existing approaches that are available?
R2: There are various other vehicle detection and classification methods based on deep learning. However, they either lack in accuracy or lack the speed of the classification. YOLO models are state-of-the-art (SOTA) in the process of multi-object detection and classification. We further improve the YOLOv3 version (SOTA of 2019) and assemble YOLOv5 (current SOTA) to address several challenges in real-time multi-vehicle classification. Also, there lacked a vehicle classification dataset that could classify the vehicles into seven classes that we classify. On top of that, we tackle one of the most prominent challenges i.e., the domain-shift problem, with transfer learning-based fine-tuning. With these advancements in the research domain and improvement of the algorithm, our method is ahead of other existing methods. We have now added a new paragraph (lines 46-68) in the introduction (section 1) to highlight the importance of our method, thanks to your valuable comment.
C3. I think that vehicle detection approaches are not really important at the present time because we already have such systems to do such tasks. But if your approach can also detect the vehicle number along with the type of vehicle then it can be a better approach. Maybe you would like to consider this in the revised version.
R3: As you mentioned, the vehicle detection problem is a well-researched problem in the past. However, the conventional methods when applied to a real-world problem such as ours fail to produce high accuracy in near real-time. Moreover, the conventional methods are more focused on the detection of vehicles, rather than classification. This has been described in paragraphs 2 and 3 of section 2.1. Also, section 2.3 provides the related recent works that continuously improved the domain of study.
Adding to the lack of robustness, the conventional methods fail in noisy surveillance videos due to changes in light conditions. This is another factor that highly affects the traditional methods. We develop a robust deep learning-based method that not only minimizes the effects of domain-shift that occurs from camera to camera but also performs well during multiple lighting conditions. The experiments that support the minimization of domain-shift are shown in section 4.2. Tables 3 and 4 respectively show the performance of the model before and after fine-tuning that was done to minimize the effects of the domain-shift. Also, section 4.4 demonstrates the experiments carried out in adverse lighting conditions.
Thank you for your suggestion to add detection of the vehicle number. Vehicle number detection aka. license plate recognition (LPR) is a different study domain and does not fall in the multi-object tracking (MOT) domain. LPR requires detection of the number plates and is followed by character recognition. Character recognition is a completely different research topic, especially when it comes to Thai character recognition due to 72 alphabet characters, some of them looking extremely alike. In summary, LPR is currently not implementable in this research paper. We are currently also carrying out LPR research separately and plan to integrate it into our MOT research works in the future. To make this clear to the readers, we have added this as future work and recommendations in the conclusion (line 600).
Round 2
Reviewer 1 Report
The authors have addressed all my comments.
Reviewer 3 Report
The authors have wisely responded to my comments. I agree with the authors. The article can be accepted.